# Applications of machine learning in behavioral ecology: Quantifying avian incubation behavior and nest conditions in relation to environmental temperature

Wayne D. Hawkins [1,2,3], Sarah E. DuRant[3]*

**1** Life Sciences Institute, University of Michigan, Ann Arbor, MI, United States of America, **2** Department of Molecular, Cellular and Developmental Biology, University of Michigan, Ann Arbor, MI, United States of America, **3** Department of Biological Sciences, University of Arkansas, Fayetteville, Arkansas, United States of America

* sedurant@uark.edu

**Data Availability Statement:** The incubation data are posted on github along with the program and source code (https://github.com/wxhawkins/NestIQ).

## Abstract

In the age of machine learning, building programs that take advantage of the speed and specificity of algorithm development can greatly aid efforts to quantify and interpret changes in animal behavior in response to abiotic environmental factors, like temperature. For both endotherms and ectotherms, temperature can affect everything from daily energy budgets to nesting behaviors. For instance, in birds environmental temperature plays a key role in shaping parental incubation behavior and temperatures experienced by embryos. Recent research indicates that temperatures experienced by embryos affect viability and are important in shaping fitness-related traits in young birds, sparking renewed interest in relationships among environmental factors, parental incubation behavior, and incubation temperature. Incubation behavior of birds can be monitored non-invasively by placing thermal probes into the nest and analyzing temperature fluctuations that occur as parents attend and leave the nest (on- and off-bouts, respectively). When other measures of temperature (e.g., ambient air or operative temperature) are collected simultaneously with incubation temperature it is possible to compare shifts in behavior with environmental changes. To improve analysis of incubation behavior using these large thermal data sets we developed a program, NestIQ, that uses machine learning to guide parameter optimization allowing it to track the behavior of diverse species. NestIQ's algorithm was tested using six species incubating in lab or field scenarios, that exhibit unique incubation patterns. This stand-alone and open source software is operated through a graphical user interface (i.e., no user programming is required) that provides important behavioral and thermal output statistics. Further, measures of environmental temperature can be imported alongside nest temperature into the program, which then reports various attributes of environmental temperature during shifts in parental behavior. This program will improve the ability of avian ecologists to interpret a critical parental care behavior that can be used across diverse incubation scenarios and species. Although specifically designed for quantifying avian incubation, NestIQ has the

**Funding:** The authors received no specific funding for this work.

**Competing interests:** The authors have declared that no competing interests exist.

potential for broader applications, including basking and nesting behaviors of non-avian reptiles in relation to ambient temperature.

## Introduction

Animals continually make fine-tuned changes to their behavior based on environmental conditions. Understanding how these decisions affect their survival, reproductive success, and subsequent interactions with their environment are essential for aiding conservation efforts in today's rapidly changing environment and for appreciating the plasticity of and constraints acting on animal behaviors. Among these rapid environmental changes include increases in average and extreme temperatures across the globe [1], which will have a tremendous affect on how animals interact with their environment [2–4]. Drastic changes in temperature averages and extremes will affect most aspects of an animal's biology, such as body temperature and thermoregulatory behaviors of ectotherms [5–7], temperatures experienced by embryos of egg-laying species [8–10], as well as behaviors in endotherms that are driven by environmental temperature, like avian incubation behavior (e.g., [11,12]).

Egg incubation in birds is both critical for proper development of embryos and taxing for the parents, necessitating a delicate balance of resource allocation to self-maintenance and offspring care. Incubation increases metabolic rates of parents while also reducing the time they can allocate to acquiring food, reducing the overall condition of these birds [13,14]. However, incubation is necessary for embryonic development and hatching success, since exposure to high or low temperatures can result in embryonic mortality [8,15]. Additionally, it is increasingly apparent that the effects of incubation temperature do not end upon hatching. In both precocial and altricial species, incubation temperature affects a suite of post-hatching traits critical to fitness of young birds (e.g., [16–21]), as well as secondary sex ratios [22,23]. These findings suggest that incubation temperature plays an important role in ecological and evolutionary processes in birds and should be an important factor considered in avian conservation efforts [10,15,17,24]. Therefore, the ability to compare incubation behavior of parents under various scenarios in relation to temperatures experienced by eggs can yield important insights into factors shaping avian reproduction.

Analyzing temperature fluctuations that eggs experience has become a standard method of quantifying avian incubation behavior, as well as thermal conditions experienced by embryos [11,16,25–27]. Using thermal data to assess parental incubation behavior provides valuable insight into the factors shaping embryonic outcomes and parental care decisions, because it provides details about the implications of an on- or off-bout, like cooling rate, relative to numerous biotic and abiotic conditions, such as ambient temperature clutch size, nest location, etc. These thermal data are relatively easy to collect; however, analyzing these data and comparing incubation decisions with ambient environmental conditions and their combined effects on nest thermal dynamics can be difficult and time-consuming. With this in mind we developed NestIQ, a stand-alone and open-source program designed specifically to detect incubation patterns while simultaneously relating these to measures of environmental temperature. Specifically, NestIQ identifies and analyzes upward and downward trends in datasets comprised of regularly interspaced values, such as thermal data collected at regular intervals. Nesting conditions and behaviors vary across species, populations, years, clutches, and treatments [9], necessitating unique algorithm settings for optimal on- and off-bout detection accuracy. NestIQ uses machine learning, a branch of artificial intelligence, to make parameter

optimization of the algorithm easier and more flexible without the need for sustained user instruction. NestIQ quickly yields a specialized algorithm tailored to the input data provided from a particular species or incubation scenario. Importantly, because the software is operated exclusively from a graphical user interface, its operation does not require scripting, or even command line familiarity. The graphical interface also allows users to quickly identify missed off-bouts to assess and refine the algorithm.

The program will also help yield insight into the connection between environmental conditions and avian incubation behavior, because when two variables are collected at identical time intervals, both of these variables can be analyzed by the program to provide various information relating one variable to the other. For instance, if air and egg temperature data are collected simultaneously, the program will provide information on ambient temperature at the time a bird leaves the nest. This ability of the program greatly enhances a biologist's ability to relate changes in adult incubation behavior as it relates to ambient temperature and other features of the clutch, nest, and parental condition. For instance, whether parental incubation decisions relative to ambient temperature shift with clutch size to effect thermal properties of the nest. To assess NestIQ's accuracy for a given system, avian biologists should validate a subset of software predictions with a more direct observation such as video recordings of incubating parents (e.g., [28]). New programs are now available that automate video analysis of nesting behavior [29,30], and could easily be partnered with analyses of thermal characterisitcs of the nest using NestIQ. Although NestIQ was designed specifically with the intent of quantifying avian incubation behavior in relation to air temperature, the program has broad applicability for other scenarios, including analyzing basking behavior in ectotherms, and assessing relationships between air and egg temperature in the nests of non-avian reptiles.

## Software/Algorithm structure

NestIQ was written in Python 3 [31] and uses a first-order hidden Markov model (HMM) to derive incubation states. HMMs assume you have a sequence of observations (also known as emissions) and that these observations are dependent on the status of an unobserved *state*. In the case of NestIQ, temperature change from the previous measurement is the observation and the two unobserved states are incubating and not incubating (on- and off-bout, respectively). HMMs consist of a collection of probabilities. The initial probabilities describe the likelihood of beginning in a given state. The transition probabilities define the likelihood of going from one state to another e.g. going from on-bout status at data point $i$ to off-bout status at data point $i$+1 (Fig 1). Lastly, the observation probabilities of NestIQ's model reflect the likelihood of observing a given temperature change in a given state; in general, the more negative the temperature change from previous, the more likely that point is to be classified as an off-bout. Because temperature change values are continuous rather than discrete, two cumulative distribution functions (one for each state) are used to calculate this probability (Fig 1). Mean and standard deviation of temperature change define the shape of these functions. In total, there are ten non-intuitive yet critical values that must be provided to the model for it to accurately assign states to the input data: two initial probabilities, four transition probabilities, two observation means and two observation standard deviations. However, users are not required to manually set any of these ten model parameters despite NestIQ giving them the ability to do so on the "Advanced" tab.

Both unsupervised and supervised approaches can be taken to automatically set NestIQ's model parameters with both having advantages in certain situations. Unsupervised learning is accomplished through the Baum-Welch algorithm. This algorithm predicts the hidden states for each data point provided, assigns a probability or score to this prediction, adjust the model

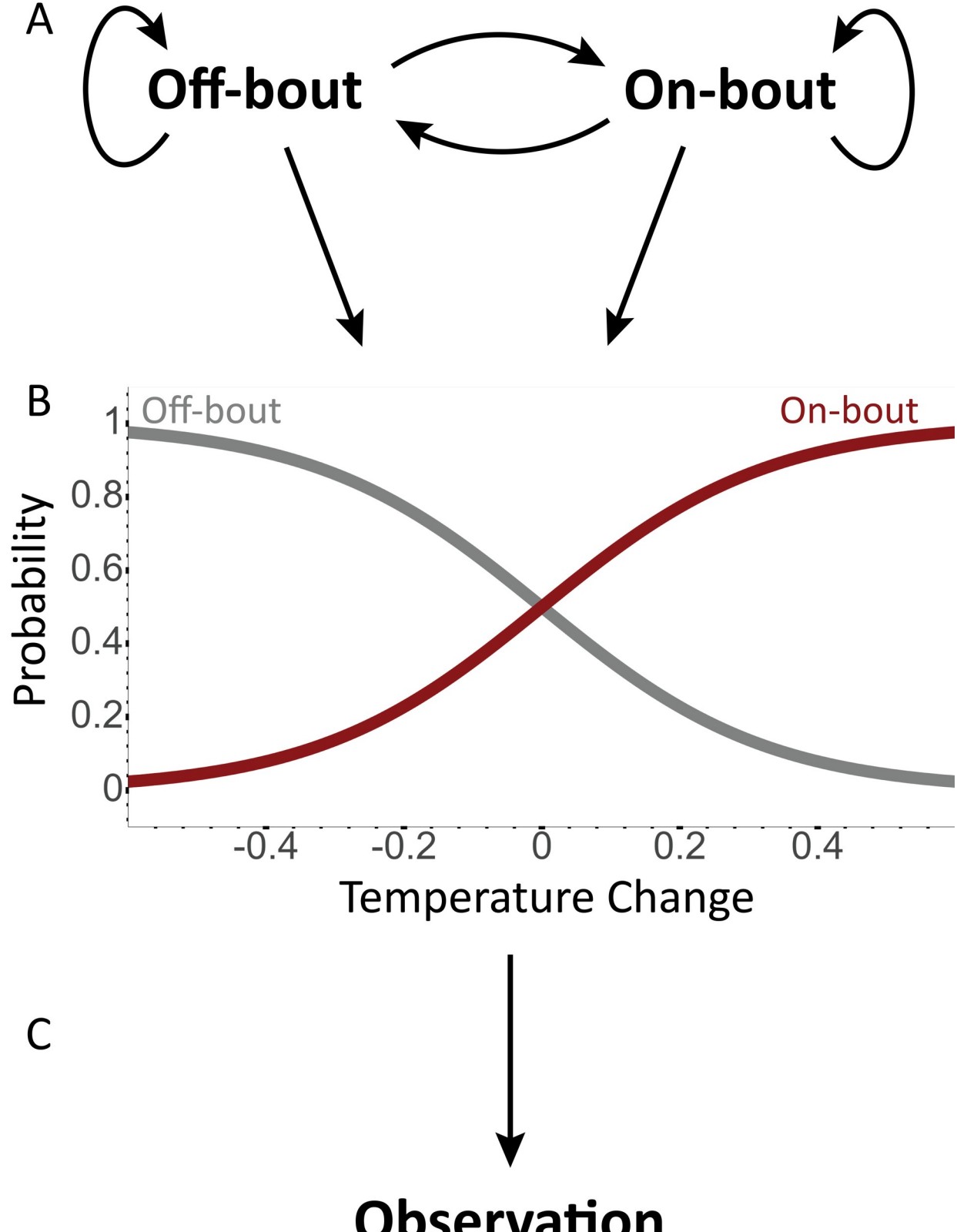

**Fig 1. Diagram of NestIQ's hidden Markov model. A.** There are two hidden states (on- and off-bout) resulting in four possible state transitions, counting transitions to the same state. Each of these is assigned a probability. **B.** The values set for the mean and standard deviation of

temperature changes in each state dictate the nature of the cumulative distribution functions. Generally, the more positive the temperature change, the higher the on-bout probability and the lower the off-bout probability. **C.** These functions are applied to a given temperature changes to yield observation probabilities that will factor into the final state assignment. *Exaggerated values are used here for demonstration purposes.*

parameters and repeats this process until the score no longer improves to a significant degree (convergence is reached). To use unsupervised learning, the user simply uploads their data and clicks the "Unsupervised Leaning" button on the "Advanced" tab. Because this approach is unsupervised, no guidance or description of the states whatsoever must be provided, only an input file containing temperature readings. When using unsupervised learning the user can set a critical threshold change in temperature and duration of change in temperature to define what constitutes an off-bout, which is input on the "Main" tab. This feature of the program is similar to programs like Rhythm [32]. This workflow is convenient but may not be as accurate as the supervised workflow and may be appropriate in special circumstances, for instance, when off-bouts are fairly similar throughout incubation.

In many cases, a supervised approach to parameter optimization is more appropriate. The supervised approach can be accessed through the "Advanced" tab. The "Select vertices" button generates a plot of the provided input data. From here the user (for a small subset of the input data) manually selects where it appears a bird is leaving or returning to the nest based on changes in nest temperature relative to ambient temperature and feeds this HTML plot file back to the software so that it may derive model parameters from the selections made. The user easily can view how well the algorithm predicted on- and off-bouts using the graphical interface where thermal data is overlaid with the algorithms defined incubation patterns. If the algorithm has underperformed it can easily be retrained. Once the algorithm meets the users standards, its parameters can then be used to analyze the rest of the provided input file as well as any other input files displaying similar incubation behavior. This approach gives the user much more control over the algorithm's behavior in assigning states. The full contents of the "Advanced" tab can be seen in Fig 2.

Optionally, the "Main" tab allows the user to "smooth" their input data. This causes NestIQ to use a rolling mean of the input data for all operations thus reducing the effect of minor fluctuation in temperature due to instrument inaccuracy or minor shifts in animal position. Lastly, NestIQ has the capability of using either raw egg temperature or the difference between egg and air temperature (adjusted egg temperature) for parameter optimization and incubation state assignment. Adjusted egg temperatures can also be plotted in place or along-side raw egg and air temperatures, offering a unique view of incubation behavior that will be useful in visualizing how egg temperature and incubation decisions relate to environmental temperature (Fig 3B).

## Input and output

A simple graphical user interface (Fig 2) was developed for NestIQ. The program takes as input one or more comma separated value (CSV) files containing columns corresponding to a data point number (e.g., 1st, 2nd, 3rd data point), date/time, egg temperature, and optionally, environmental temperature (e.g., ambient air temperature; Fig 4). NestIQ supports the selection of multiple input files, in other words, the program can analyze thermal data acquired from multiple nests simultaneously. This ability allows the user to generate individual plots and statistics summaries for each nest and, importantly, enables the automatic calculation of numerous compiled statistics to aid the identification of incubation patterns across all nests (discussed in detail below).

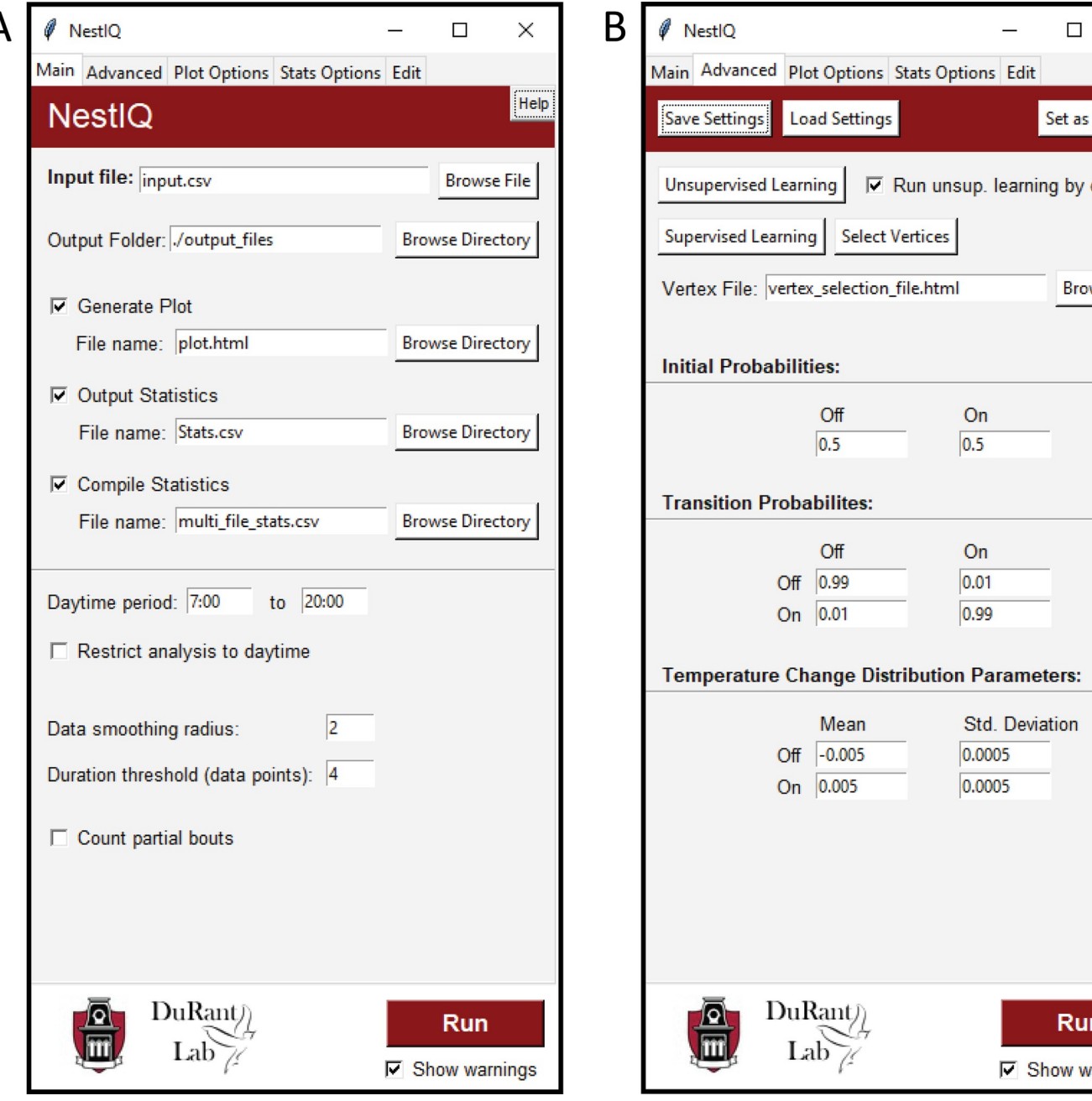

**Fig 2. NestIQ's graphical user interface. A.** The Main tab allows for changing input/output settings and file names as well as a few important program parameters. **B.** The Advanced tab is where machine learning functionalities and core model parameters are housed.

NestIQ offers multiple output options for visualizing the algorithm-generated on- and off-bout predictions and creating files that report relevant statistics. For each thermal profile provided, the "Generate plot" option creates an interactive plot (Figs 3 and 5) through the Python module, Bokeh [33]. The "Plot Options" tab was implemented to allow for configurability in the plot's appearance. NestIQ-generated plots also have several functions that allow for easy maneuverability of large thermal datasets from single nests, including panning via dragging with the mouse, box zoom, and hover info (Fig 5). The "Output statistics" option provides

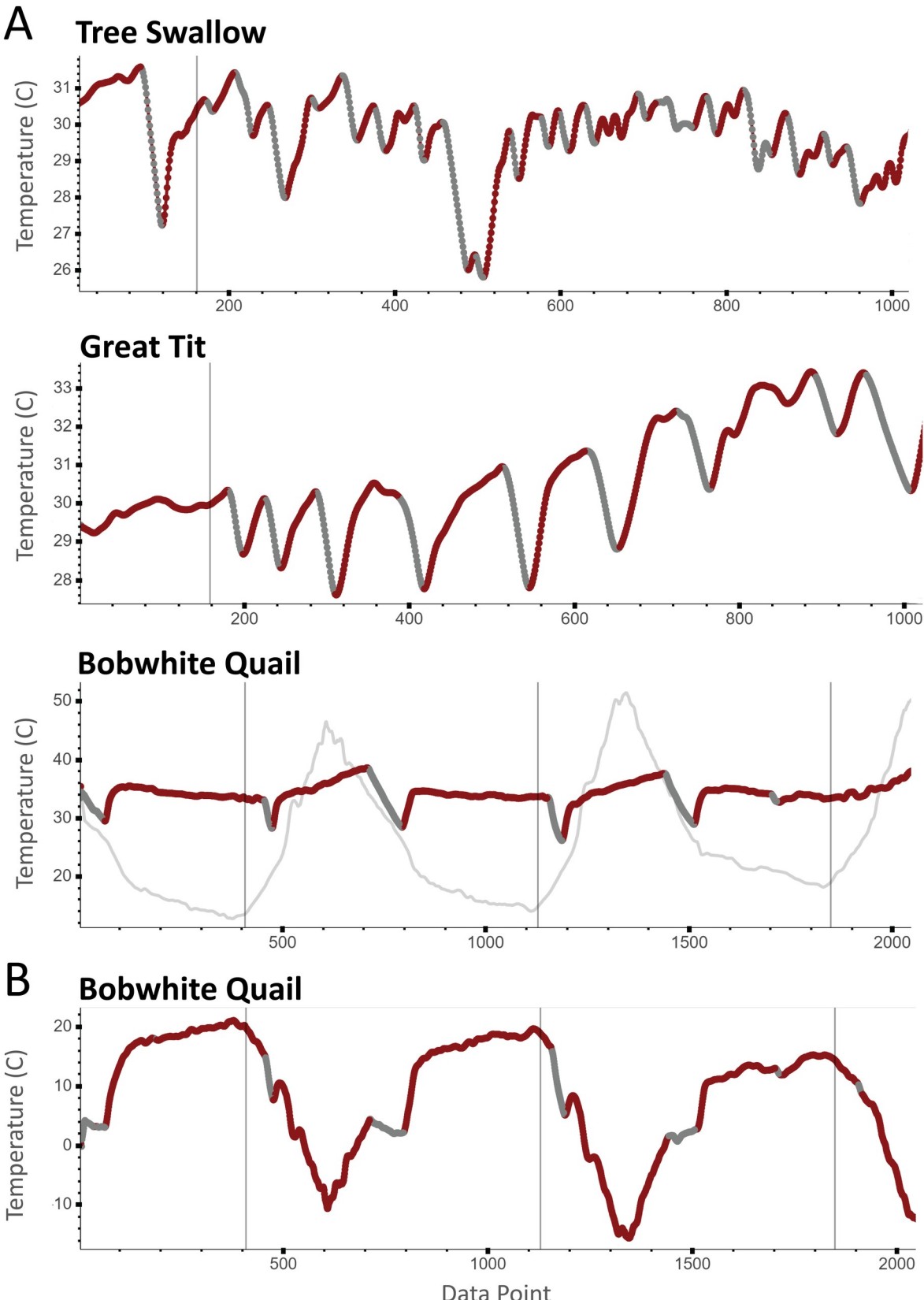

**Fig 3. Bout prediction results for incubation data from three species.** NestIQ was used to analyze and plot data from Tree Swallow (*Tachycineta bicolor*), Great Tit (*Parus major*), and Bobwhite Quail (*Colinus virginianus*). Vertical lines in each plot mark 7:00 AM. Temperature measurements were collected every 30 seconds for great tits and tree swallows and every two minutes for bobwhite quail. For each plot, a rolling mean was used to smooth the data. **A.** Raw egg and air temperatures are plotted with predicted on-bouts shown in red and off-bouts in gray. The light gray line shows air temperature. Note, air temperature cannot be seen in the tree swallow and great tit plots because it was much lower than nest temperature. Zooming out would have obscured minor fluctuations in incubation patterns. The input files and corresponding configuration files are included with NestIQ distributions as examples. **B.** Adjusted egg temperatures (egg temperature minus air temperature) are plotted for the same bobwhite quail data shown in Fig 3A. Bout predictions were also unchanged between these two plots.

information about each individual bout such as duration, mean egg temperature, mean air temperature, and egg temperature change during on- and off-bouts. Additionally, at the top of this file is a summary providing an array of information and statistics for individual days of incubation as well the entire period of incubation (i.e., the entire input file). These summary statistics allow the user to assess temporal patterns in incubation behaviors (e.g. do off-bouts increase in duration as incubation progresses?) and characteristics of a nesting attempt as a whole. These characteristics can then be compared against incubation patterns of other birds. Lastly, the "Compile statistics" option outputs the same summary described above; however, if multiple input files are uploaded, statistical summaries are provided for each input file and all input files compiled. This allows for the identification of broader incubation patterns across populations or treatments. For instance, the incubation statistics for a multi-nest treatment group can be easily compiled and compared to the nesting pattern of a different treatment group (e.g., average off-bout duration of parents in disturbed vs. undisturbed habitat). This feature would also be useful for generating incubation statistics for a single female over several nesting attempts. A full list of the statistics/information reported by NestIQ can be seen in Table 1.

## Materials and methods

This software was developed to quantify avian incubation behavior, and thus was tested with and designed around datasets acquired from placing egg-resembling temperature probes in the nests of actively incubating birds following the methods of Coe *et al.* [12]. Initial thermal data used for testing were collected from canary nests using a modified HOBO Pro v2 2x External Temperature Data Logger (Part # U23-003). The thermal probe was modified with a

| Data Point | Date/Time | Egg Temperature | Air Temperature (optional) |
|---|---|---|---|
| 1 | 6/5/2020 15:00 | 32.355 | 24.098 |
| 2 | 6/5/2020 15:01 | 32.381 | 24.146 |
| 3 | 6/5/2020 15:02 | 32.381 | 24.195 |
| 4 | 6/5/2020 15:03 | 32.355 | 24.243 |
| 5 | 6/5/2020 15:04 | 32.33 | 24.291 |
| 6 | 6/5/2020 15:05 | 32.33 | 24.339 |
| 7 | 6/5/2020 15:06 | 32.33 | 24.388 |
| 8 | 6/5/2020 15:07 | 32.355 | 24.46 |
| 9 | 6/5/2020 15:08 | 32.355 | 24.508 |

**Fig 4. Example input file.**

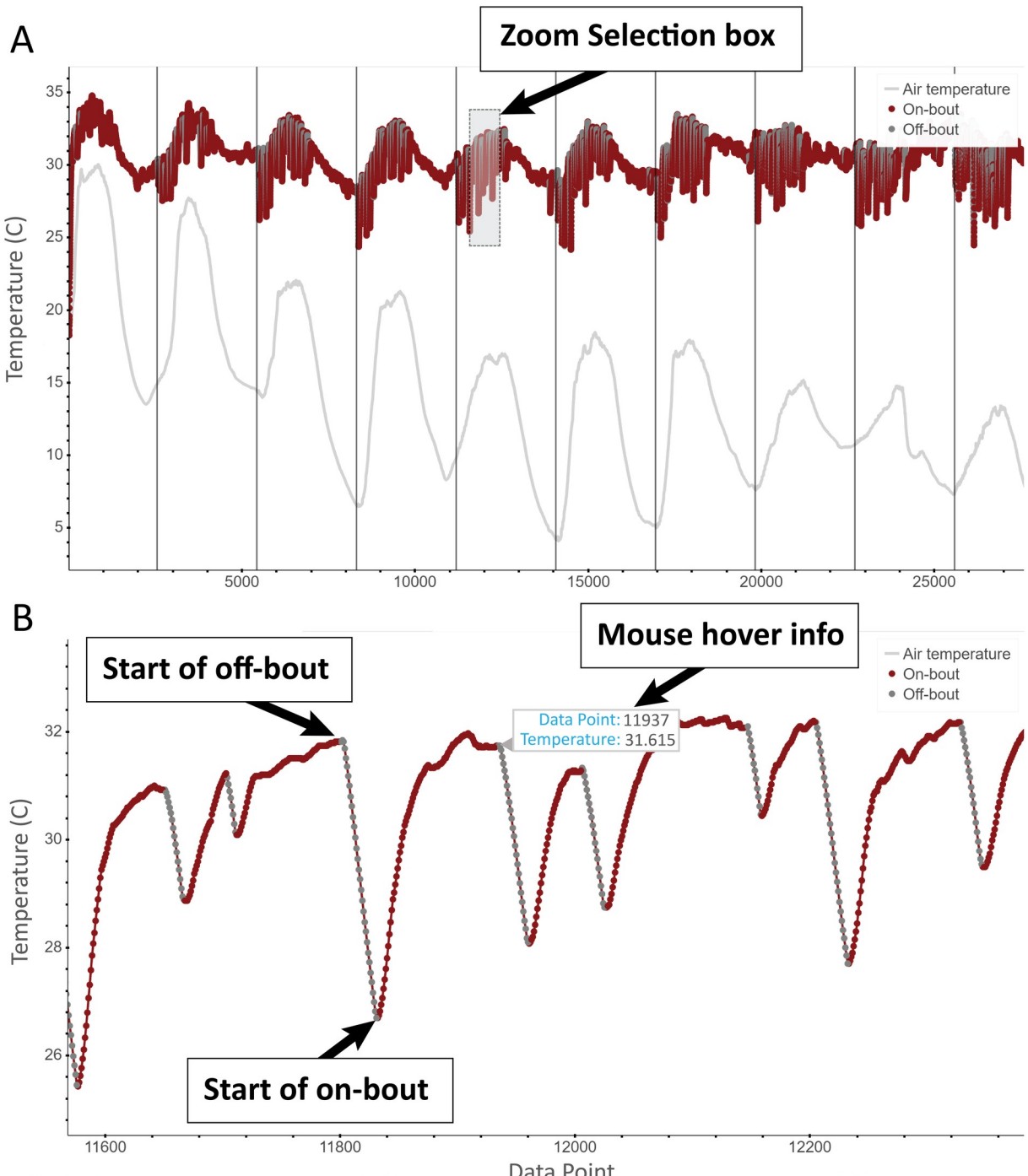

**Fig 5. Interactive plot produced from the "Generate plot" option. A.** Plot of Great Tit nest temperature fluctuation over roughly ten days with predicted on- and off-bouts. **B.** Result of zoom selection box shown in **4A**.

thermal sensor that slips inside a clay-molded egg of canary dimensions (see [12]) and was inserted up into the nest cup by feeding it through the bottom of the nest. This allows wires to run underneath the nest and outside of the cage and keeps them out of the way of incubating parents. Installment of the probes was performed on the day the first egg was laid. Loggers

**Table 1. Statistics and other information reported by NestIQ.**

| Individual Bout | Individual Day/File | Multiple files |
|---|---|---|
| Date | Bout number (on/off) | Bout number (on/off) |
| Bout type | Mean bout duration (on/off, stdev) | Mean bout duration (on/off, stdev) |
| Start time | Mean egg temperature change (on/off, stdev) | Mean egg temperature change (on/off, stdev) |
| End time | Sum of bout type time (on/off) | Number of full days |
| Start data point | Time above critical egg temperature | Mean egg temperature (stdev, day/night) |
| End data point | Time below critical egg temperature | Minimum egg temperature |
| Duration | Bouts discarded | Maximum egg temperature |
| Egg temperature change | Mean egg temperature (stdev, day/night) | Mean air temperature |
| Starting egg temperature | Median egg temperature (day/night) | Minimum air temperature |
| Ending egg temperature | Minimum egg temperature (day/night) | Maximum air temperature |
| Mean egg temperature | Maximum egg temperature (day/night) | |
| Starting air temperature | Mean air temperature (day/night) | |
| Ending air temperature | Minimum air temperature | |
| Mean air temperature | Maximum air temperature | |

**On/off:** this information is reported individually for both on- and off-bouts, **stdev:** the standard deviation of this value is also reported, **day/night:** this information is reported individually for daytime, nighttime, and combined.

were programmed to record every 30 seconds, necessitating data collection once a week, which was done wirelessly via a HOBO waterproof shuttle. The 20 canary thermal profiles used for initial alpha-testing were yielded from probes installed in one or more nesting attempts of ten females. Once the major components of the software were developed, we fine-tuned the program to ensure its effectiveness across diverse species, both altricial and preco-cial, using thermal profiles collected as described above for Great Tits (*Parus major*), Scaled Quail (*Callipepla squamata*), Bobwhite Quail (*Colinus virginianus*), Wood Ducks (*Aix sponsa*), and Tree Swallows (*Tachycineta bicolor)* nesting in natural settings. Thermal loggers were pro-grammed to log data every 2 minutes in Wood Ducks and the two quail species. Thermal pro-files from the five species were tested extensively and drove several subtle changes in the bout detection algorithm to make the program relevant for a wide range of species and incubation scenarios (Fig 3). Data used in making this program were from research approved by the IACUC committees of Virginia Tech, Oklahoma State University, and University of Arkansas.

## Limitations

Like most biological behaviors, incubation can be highly variable. Consequently, NestIQ will not always perfectly predict animal behavior. It would be ideal for users of NestIQ to validate prediction accuracy by comparing a subset of software predictions with a more direct observa-tion such as video recordings of incubating parents (e.g., [28]). In fact, NestIQ's usefulness may be greatest when used in conjunction with other programs designed to understand incu-bation in birds. For instance, BirdBox and a deep learning program can be used to analyze video observations of incubation patterns [29,30]. Like NestIQ, these programs make it possi-ble to easily interpret large datasets of incubation behavior and could be used to verify timing and duration of on- and off-bouts, while Nest IQ provides a comprehensive view of the ther-mal implications of parental decisions for developing embryos and the plasticity of parental behavior to ambient air temperatures.

We had visual observations from 3 bobwhite or scaled quail nests (observations made every thirty minutes) and our program accurately detected all of the off-bouts at these nests (n = 5 off-bouts; Carroll unpub data) and the duration of the off-bouts fit within the crude bout-lengths identified by nest observations. We did not have video or observer recorded off-bouts for any of the other species.

## Conclusions

NestIQ offers numerous features that will improve our ability to relate changes in incubation behavior with its effects on nest thermal dynamics and offspring outcomes. The program will also help researchers predict behavioral changes in animals that will occur as a result of changing environmental conditions. Useful features include output statistics on characteristics of individual bouts, like the air temperature at which birds leave and return to the nest, as well as a summary of the total time eggs were above or below a given temperature. The program also reports statistics within a user-defined time range to compare, for instance, overnight versus daytime characteristics of nests, and can process multiple input files simultaneously to calculate cumulative statistics about groups of incubating birds. Finally, the maneuverable interactive plot makes it easy to visualize and refine how well the algorithm fits an incubation profile. Although, this software was developed around avian incubation data, the use of NestIQ likely extends to many other scenarios where animal behavior and physiology are responding to environmental cues, such as analyzing fluctuations in body temperature of ectotherms.

The importance of incubation behavior and temperature to the phenotype of young birds continues to yield exciting new research avenues. Research in this area indicates that slight changes in incubation patterns could have consequences for avian reproductive success that manifest at post-hatch life stages [10,17]. Therefore, better understanding the factors that shape and disrupt normal incubation patterns will contribute to avian conservation efforts. It is critical that the tools ecologist use to assess animal behavior are continually updated to include the latest advancements in technology to better meet the evolving needs of researchers. Nest IQ hopes to meet some of these needs by improving techniques to explore the relationships between ambient temperature, parental behavior, clutch and nest attributes, and thermal dynamics of the nest.

## Acknowledgments

We thank Andy Alverson, Jeremy Beaulieu, and Krti Tallam for their input in software development. Many thanks to Sydney Hope, Brittney Coe, William Hopkins, Rachel Carroll, Craig Davis, and Sam Fuhlendorf for sharing thermal datasets from Tree Swallows, Great Tits, Scaled Quail, and Bobwhite Quail.

The software for NestIQ was written by W.D.H. The design of NestIQ was conceived by S. E.D. and W.D.H. This paper was written by S.E.D. and W.D.H.

## Author Contributions

**Conceptualization:** Wayne D. Hawkins, Sarah E. DuRant.

**Data curation:** Wayne D. Hawkins.

**Funding acquisition:** Sarah E. DuRant.

**Methodology:** Wayne D. Hawkins, Sarah E. DuRant.

**Software:** Wayne D. Hawkins.

**Supervision:** Sarah E. DuRant.

**Validation:** Wayne D. Hawkins.

**Visualization:** Wayne D. Hawkins.

**Writing – original draft:** Wayne D. Hawkins.

**Writing – review & editing:** Wayne D. Hawkins, Sarah E. DuRant.

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
