## [Decision Letter · Decision Letter 0]

3 Feb 2020

PONE-D-19-30811

Applications of machine learning in behavioral ecology: Quantifying avian incubation behavior and nest conditions in relation to environmental temperature

PLOS ONE

Dear Dr. DuRant,

Thank you for submitting your manuscript to PLOS ONE. After careful consideration, we feel that it has merit but does not fully meet PLOS ONE’s publication criteria as it currently stands. Therefore, we invite you to submit a revised version of the manuscript that addresses the points raised during the review process.

All three reviewers acknowledge the importance of your work and scientific merit. However, they also raise some valid technical/methodological concerns, which have to be addressed in a revised version.

We would appreciate receiving your revised manuscript by Mar 19 2020 11:59PM. To enhance the reproducibility of your results, we recommend that if applicable you deposit your laboratory protocols in protocols.io, where a protocol can be assigned its own identifier (DOI) such that it can be cited independently in the future. For instructions see: http://journals.plos.org/plosone/s/submission-guidelines#loc-laboratory-protocols

We look forward to receiving your revised manuscript.

Kind regards,

Roland Bouffanais, Ph.D.

Academic Editor

PLOS ONE

Reviewers' comments:

Reviewer's Responses to Questions

**Comments to the Author**

1. Is the manuscript technically sound, and do the data support the conclusions?

Reviewer #1: Yes

Reviewer #2: Partly

Reviewer #3: Yes

2. Has the statistical analysis been performed appropriately and rigorously? 

Reviewer #1: Yes

Reviewer #2: N/A

Reviewer #3: Yes

3. Have the authors made all data underlying the findings in their manuscript fully available?

Reviewer #1: Yes

Reviewer #2: Yes

Reviewer #3: Yes

4. Is the manuscript presented in an intelligible fashion and written in standard English?

Reviewer #1: Yes

Reviewer #2: Yes

Reviewer #3: Yes

5. Review Comments to the Author

Reviewer #1: Your effort to produce and share this open source program are greatly appreciated. This can be of great service to the scientific community. The comments in this review are intended in the spirit of constructive critique.

L26 I would avoid the word prove from a simple philosophy of science perspective.

L32 Typo

The article could be shortened by dropping L37-41 and moving up the information about temperature to be followed by L:43-47.

L70 Do you mean nest or egg temperature, which could differ. It would be helpful to be clear throughout, especially given variation among eggs within a nest and parent-egg ‘conflict’.

L79 citation here could be useful

The intro went quickly from the conceptual importance to the solution but without much on the ‘problem’, which is not just how to efficiently process data. Rather the ‘problem’ that your program will greatly aid researchers in solving is classifying an incubation bout given the complex nature of scoring an incubation bout ---for example, the temperature profile revealing the egg cooling rate can vary depending on ambient, clutch size, length of off bout, female size, embryo age, and so on, confounding a simple assessment of whether a bird left. This is the real issue that needs to be solved, for which it seems like your program can help considerably. So, the introduction should make this more clear, otherwise, the program might seem to the uninitiated like an efficient way to process data. In other words, help a reader see how you address both the biological and computational complexity needs to be framed and explained so that the solution proposed can be examined in context. It would also be helpful to be clear about the assumptions that are being made—not that those assumptions are bad (e.g. eggs cool when the parent leaves, eggs drop in temperature with a particular slope, etc.), but rather they need to be stated.

I’m a bit concerned about the idea that no additional information is needed from a user and that the program will use machine learning to correctly identify incubation bouts from any data set. Our experience from working with either a single species or comparing across closely-related species is that to be biologically confident two things are needed: (1) knowledge of actual incubation behavior in order to assess what an on-bout looks like in the temperature data (how much of a change in temp; what is the range of slopes) and some iterative optimization of parameters to assess error rate in order to find the thresholds of slope and/or temp change than minimize error rate. Perhaps this approach is implied in the machine learning, at the least the second step of iteratively adjusting parameters. This should be made explicit—in particular, how is the program optimizing? Minimizing sums of squares? What is the process involved.

L100 As most of the interested parties are likely to be biologists, it would be useful to give a description of the terms involved, such as ‘emissions’. Can you describe the basics of a Markov model? Right now the writing serves to demonstrate programming expertise without explaining the underlying logic of the program to a practitioner.

L104-5 As a potential user of this program, I would love to know what those critical values are so I can assess whether NestIQ will give me reliable and defensible assessments of incubation behavior.

L110 This feels like the key aspect—if the program simply sorts observations into two states (on vs. off-bout) then how often is it correct?

L114 What is “generally quite accurate”—most researchers would like to know error rate, particularly how it might vary by species—larger species with larger eggs will have slower cooling rates, so that when a bird leaves for a short time, the program will consider this still an on-bout. But from a behavioral perspective the bird has left. I’m sure error rate is low, but more details are needed.

Reviewer #2: The authors provide a description of a new software – NestIQ – which aims to quantify avian incubation patterns (on- and off-bouts) and enables linking incubation data to environmental conditions. This software could certainly be a useful tool for scientists studying incubation patterns, particularly due to the lack of programming needed to use it and the simple data which it takes as inputs. The paper is generally clear and well written and represents a useful scientific contribution.

Main comments

1) The manuscript is lacking a discussion/acknowledgement of other similar software/methods currently in use to analyze incubation patterns. I suggest adding a paragraph of background on this to the introduction. For example, Amininasab et al 2016 introduce BirdBox software to analyze video data to document incubation, and Williams and DeLeon 2020 use deep learning to analyze incubation

2) The authors refine the model using data from several very different species with different nest types. This is great, but it would be interesting to know a little more detail about how/if the different nesting strategies affected the quality of model output. E.g. Does the model need to me made more sensitive to recognize off bouts for cavity nesters as opposed to open nest species, where the temperature change is presumably not as big/quick.

3) My main reservation with this paper is the lack of any clearly described validation step, although the authors do point out that users would ideally validate accuracy with visual data (line 191-194). Is there any data the authors can add to let the reader know how accurate the software is for the species they tested it on? I strongly suggest that this be added if at all possible, even if only for a subset of the species studied. Without this it is impossible to know if this is a useful tool.

4) The software separates out day and night incubation statistics which is a great feature. Can the authors add a line to clarify whether the user defines start and end times for day and night, or if this is an inbuilt feature?

Minor comments

Line 49 – 51 – These statements need references.

Line 57 - Williams et al. 1996 is cited in the text but is missing from the reference list.

Line 60 – Cooper et al 2005 could refer to one of two papers in the references. Please clarify with an a or b.

Line 100 – The term ‘emissions’ in this context is not instantly clear to me. I suggest either rephrasing this using more commonly used terms, or defining the exact usage of ‘emissions’.

Line 104 – What are there 10 critical values? Is this something the user has to input? Please clarify this statement.

Line 108 – Should there be a reference for the python package?

Line 150 – Should there be a reference for the python package?

Line 159 – Missing a question mark at the end of the example in parentheses

Line 172 – Coe et al 2015 is cited in the text but missing from the reference list.

Line 186 – unnecessary period after Tachycineta bicolor.

Line 215 – Needs a reference

Figures – all the figures are very pixelated and poorly rendered in my version, making it hard to read small text and the GUI figure, so this has not formed part of my review. This will need to be checked and higher resolution images provided (if it is not simply an artefact of the manuscript upload system)

References – I suggested cross checking the cited references and the reference list as I found two missing citations just from a quick glance through the list.

References:

Amininasab SM, Kingma SA, Birker M, Hildenbrandt H, Komdeur J (2016) The effect of ambient temperature, habitat quality and indi- vidual age on incubation behaviour and incubation feeding in a socially monogamous songbird. Behav Ecol Sociobiol 70:1591– 1600

Williams, H.M. and DeLeon, R.L. (2020) Deep learning analysis of nest camera video recordings reveals temperature-sensitive incubation behavior in the purple martin (Progne subis). Behavioral Ecology and Sociobiology 74:7 https://doi.org/10.1007/s00265-019-2789-2

Reviewer #3: The basic work and ideas seem sound and well thought-out. The NestIQ utility may be of use to some researchers studying processes such as incubation. The authors describe the use cases and provide a reasonable amount of information on the algorithm variables, output and operation of the NestIQ program.

While the basic implementation of the supervised vs unsupervised learning was described it was not indicated when or under what conditions supervised learning would be an improvement over unsupervised learning. It would be better to inform the reader what data he would need for supervised learning and how it would improve the model. As it is, there is no guidance given and no idea what the actual value that the supervised learning provides. While the way to use smoothing and thresholding are more intuitive and probably do not need any further explanation, the authors similarity do not indicate why one would make manual adjustments to the model parameters or how this could be advantageous.

The authors provided a link to the NestIQ github repository. I followed this but I was not able to get NestIQ to run. While someone more motivated would undoubtedly be able to work through the the additional package installations and sort through the run errors that frustrated my efforts, it does highlight the need for better installation/operation instructions in the manual. Additionally, installation could be described in the paper itself.

6. PLOS authors have the option to publish the peer review history of their article (what does this mean?). If published, this will include your full peer review and any attached files.

Reviewer #1: No

Reviewer #2: No

Reviewer #3: No

---

## [Author Response · Author response to Decision Letter 0]

7 Jul 2020

We thank the reviewers for their insightful comments, which improved the manuscript significantly.

Reviewer #1: 

Your effort to produce and share this open source program are greatly appreciated. This can be of great service to the scientific community. The comments in this review are intended in the spirit of constructive critique.

L26 I would avoid the word prove from a simple philosophy of science perspective.

We’ve made edits to the text to delete the use of prove

L32 Typo

This was corrected.

The article could be shortened by dropping L37-41 and moving up the information about temperature to be followed by L:43-47.

We made these suggested changes.

L70 Do you mean nest or egg temperature, which could differ. It would be helpful to be clear throughout, especially given variation among eggs within a nest and parent-egg ‘conflict’.

We appreciate the reviewers concern and have made edits to the text to be more broad, since variations in both egg and nest temperatures can be used to determine adult incubation behavior. 

L79 citation here could be useful

We’ve added citations here for variable incubation patterns across species.

The intro went quickly from the conceptual importance to the solution but without much on the ‘problem’, which is not just how to efficiently process data. Rather the ‘problem’ that your program will greatly aid researchers in solving is classifying an incubation bout given the complex nature of scoring an incubation bout ---for example, the temperature profile revealing the egg cooling rate can vary depending on ambient, clutch size, length of off bout, female size, embryo age, and so on, confounding a simple assessment of whether a bird left. This is the real issue that needs to be solved, for which it seems like your program can help considerably. So, the introduction should make this more clear, otherwise, the program might seem to the uninitiated like an efficient way to process data. In other words, help a reader see how you address both the biological and computational complexity needs to be framed and explained so that the solution proposed can be examined in context. It would also be helpful to be clear about the assumptions that are being made—not that those assumptions are bad (e.g. eggs cool when the parent leaves, eggs drop in temperature with a particular slope, etc.), but rather they need to be stated.

We thank the reviewer for this conceptual comment and have made edits throughout the introduction to highlight the value of the program beyond its ability to “crunch” data (ln 41-48, 99-102, 140-144).

I’m a bit concerned about the idea that no additional information is needed from a user and that the program will use machine learning to correctly identify incubation bouts from any data set. Our experience from working with either a single species or comparing across closely-related species is that to be biologically confident two things are needed: (1) knowledge of actual incubation behavior in order to assess what an on-bout looks like in the temperature data (how much of a change in temp; what is the range of slopes) and some iterative optimization of parameters to assess error rate in order to find the thresholds of slope and/or temp change than minimize error rate. Perhaps this approach is implied in the machine learning, at the least the second step of iteratively adjusting parameters. This should be made explicit—in particular, how is the program optimizing? Minimizing sums of squares? What is the process involved.

Thank you for pointing out this concern. We’ve made edits to the text to make it clear that in the supervised learning, the user still trains the algorithm, by identifying on and off bouts, which the program then uses to learn that individual’s incubation pattern (lines 132-134, 143-160, 243-311). After using that training to detect on/off bouts in a dataset the user can still modify on/off bouts set by the algorithm. This can be done easily by viewing the thermal data overlaid with the algorithms defined incubation patterns, and manually changing where an on/off bout occurs. Viewing the algorithm plots on top of the raw thermal data make it very quick and easy to identify places where the algorithm underperformed, and allows the user to better retrain the algorithm. In other words the user still sets the guidelines for what should or should not be considered an off-bout, then the program takes over. As pointed out in the manuscript, ideally, biologists will use video observations to confirm on- and off-bouts in a subset of nests (ln 143-160, 376-392). 

We’ve also added text to make it clear that the unsupervised learning functions requires the user to provide threshold changes in temperature and duration of temperature change to allow the program to identify on-and off-bouts (Ln 229-242). This feature is similar to exisiting programs like Rhythm and Raven. Similar to the algorithm training, the user will need knowledge of actual incubation behavior to set these thresholds. 

L100 As most of the interested parties are likely to be biologists, it would be useful to give a description of the terms involved, such as ‘emissions’. Can you describe the basics of a Markov model? Right now the writing serves to demonstrate programming expertise without explaining the underlying logic of the program to a practitioner.

We are excited to see your interest on this topic. We were unsure to what length we should describe the model and algorithms in the manuscript but have now expanded on this topic in the “Software/Algorithm Structure” section. Additionally, the term “emissions” has been substituted for the hopefully more intuitive term “observations” and described further in the previously mentioned section of the manuscript.

L104-5 As a potential user of this program, I would love to know what those critical values are so I can assess whether NestIQ will give me reliable and defensible assessments of incubation behavior.

This has now been elaborated on in the “Software/Algorithm Structure” section of the manuscript (Ln 225-228).

L110 This feels like the key aspect—if the program simply sorts observations into two states (on vs. off-bout) then how often is it correct?

Training of the algorithm will vary with species and incubation context. The ability of the algorithm to detect on- and off-bouts across species depends primarily on the person training the algorithm, specifically the thresholds they use for defining an on-and off-bout. The more difficult environment for detecting on and off-bouts is when air temperatures and nest/egg temperatures are similar—this isn’t a failure of the program, however, but a drawback of using air and nest thermal data to quantify incubation behavior.

L114 What is “generally quite accurate”—most researchers would like to know error rate, particularly how it might vary by species—larger species with larger eggs will have slower cooling rates, so that when a bird leaves for a short time, the program will consider this still an on-bout. But from a behavioral perspective the bird has left. I’m sure error rate is low, but more details are needed.

We’ve edited this statement. The accuracy of the algorithm will depend on training of the algorithm. The user will determine what thresholds dictate an on- or off-bout and use those criteria to train the algorithm. Manual edits to the algorithms output can still be made using the interactive graphs produced by the program, but users should have knowledge of the incubation behavior of the species they are analyzing. We’ve also added the limited data we have that verify NestIQ identification of off-bouts and off-bout duration with visual observations (Lines 407-411; described in detail below).

Reviewer #2: 

The authors provide a description of a new software – NestIQ – which aims to quantify avian incubation patterns (on- and off-bouts) and enables linking incubation data to environmental conditions. This software could certainly be a useful tool for scientists studying incubation patterns, particularly due to the lack of programming needed to use it and the simple data which it takes as inputs. The paper is generally clear and well written and represents a useful scientific contribution.

Main comments

1) The manuscript is lacking a discussion/acknowledgement of other similar software/methods currently in use to analyze incubation patterns. I suggest adding a paragraph of background on this to the introduction. For example, Amininasab et al 2016 introduce BirdBox software to analyze video data to document incubation, and Williams and DeLeon 2020 use deep learning to analyze incubation.

Thank you for drawing our attention to these journal articles. We’ve added discussion of these programs to the manuscript (ln:143-160, 400-406).

2) The authors refine the model using data from several very different species with different nest types. This is great, but it would be interesting to know a little more detail about how/if the different nesting strategies affected the quality of model output. E.g. Does the model need to me made more sensitive to recognize off bouts for cavity nesters as opposed to open nest species, where the temperature change is presumably not as big/quick.

This is an excellent point. Training of the algorithm will vary with species and incubation context. The ability of the algorithm to detect on- and off-bouts across species depends primarily on the person training the algorithm, specifically the thresholds they use for defining an on-and off-bout. The more difficult environment for detecting on and off-bouts is when air temperatures and nest/egg temperatures are similar—this isn’t a failure of the program, however, but a drawback of using air and nest thermal data to quantify incubation behavior.

3) My main reservation with this paper is the lack of any clearly described validation step, although the authors do point out that users would ideally validate accuracy with visual data (line 191-194). Is there any data the authors can add to let the reader know how accurate the software is for the species they tested it on? I strongly suggest that this be added if at all possible, even if only for a subset of the species studied. Without this it is impossible to know if this is a useful tool.

We have visual observations on a small subset of the quail nests. The observer and NestIQ detected the same off-bouts and off-bout durations; however, the observer durations were crude because observations were made every 30 minutes (Line 407-411). However, it’s very clear in the nest thermal figures when parents leave and return to the nest, and match published reports on the timing of on- and off-bouts for species used in this study (e.g., Carroll et al. 2018; Coe et al. 2015; Hope et al. 2018; 2020). We agree with the reviewer, though, that video observations will be critical in verifying timing and duration of off-bouts in scenarios when ambient and nest temperatures are similar.

4) The software separates out day and night incubation statistics which is a great feature. Can the authors add a line to clarify whether the user defines start and end times for day and night, or if this is an inbuilt feature?

We’ve added text to address this (Ln 423-425). Yes, the author defines start and end times. So, although this feature was designed to compare night and day patterns, it can be used to quantify incubation patterns within any timeframe relevant to the biologist. 

Minor comments

Line 49 – 51 – These statements need references.

We’ve added references

Line 57 - Williams et al. 1996 is cited in the text but is missing from the reference list.

We’ve made this edit.

Line 60 – Cooper et al 2005 could refer to one of two papers in the references. Please clarify with an a or b.

We’ve made this edit.

Line 100 – The term ‘emissions’ in this context is not instantly clear to me. I suggest either rephrasing this using more commonly used terms, or defining the exact usage of ‘emissions’.

“Emissions” has been substituted with “observations” and more thoroughly described (line 167-168).

Line 104 – What are the 10 critical values? Is this something the user has to input? Please clarify this statement.

These are now listed and followed by an explicit note that the user is not expected to manually set these. (line 179-228).

Line 108 – Should there be a reference for the python package?

This reference has been added.

Line 150 – Should there be a reference for the python package?

This reference has been added.

Line 159 – Missing a question mark at the end of the example in parentheses

We’ve made this edit.

Line 172 – Coe et al 2015 is cited in the text but missing from the reference list.

We’ve added this citation to the references.

Line 186 – unnecessary period after Tachycineta bicolor.

We’ve made this edit.

Line 215 – Needs a reference

We’ve made this edit

Figures – all the figures are very pixelated and poorly rendered in my version, making it hard to read small text and the GUI figure, so this has not formed part of my review. This will need to be checked and higher resolution images provided (if it is not simply an artefact of the manuscript upload system)

Thank you for pointing this out. We will ensure high resolution figures are used in the publication.

References – I suggested cross checking the cited references and the reference list as I found two missing citations just from a quick glance through the list.

Thanks for catching this error. We’ve crosschecked references and citations. 

References:

Amininasab SM, Kingma SA, Birker M, Hildenbrandt H, Komdeur J (2016) The effect of ambient temperature, habitat quality and individual age on incubation behaviour and incubation feeding in a socially monogamous songbird. Behav Ecol Sociobiol 70:1591– 1600

Williams, H.M. and DeLeon, R.L. (2020) Deep learning analysis of nest camera video recordings reveals temperature-sensitive incubation behavior in the purple martin (Progne subis). Behavioral Ecology and Sociobiology 74:7 https://doi.org/10.1007/s00265-019-2789-2

Reviewer #3: 

The basic work and ideas seem sound and well thought-out. The NestIQ utility may be of use to some researchers studying processes such as incubation. The authors describe the use cases and provide a reasonable amount of information on the algorithm variables, output and operation of the NestIQ program.

While the basic implementation of the supervised vs unsupervised learning was described it was not indicated when or under what conditions supervised learning would be an improvement over unsupervised learning. It would be better to inform the reader what data he would need for supervised learning and how it would improve the model. As it is, there is no guidance given and no idea what the actual value that the supervised learning provides. While the way to use smoothing and thresholding are more intuitive and probably do not need any further explanation, the authors similarity do not indicate why one would make manual adjustments to the model parameters or how this could be advantageous.

The manuscript has now been updated to better convey when and how to use the unsupervised vs supervised learning workflow. In brief, unsupervised learning requires minimal time investment from the user and thus can allow for “quick and dirty” assessment. When more precise analysis is needed, the user can use the supervised learning method to guide the algorithm. How this is done is now more thoroughly covered in the text.

The authors provided a link to the NestIQ github repository. I followed this but I was not able to get NestIQ to run. While someone more motivated would undoubtedly be able to work through the additional package installations and sort through the run errors that frustrated my efforts, it does highlight the need for better installation/operation instructions in the manual. Additionally, installation could be described in the paper itself.

We apologize for this and recognize that this could have been the result of a true bug. We have performed additional error checking in the time since our initial submission. Unfortunately, the program currently only works on Windows, not MacOS or Linux which may be one reason the program didn’t work for you. We’ve now made this clear in the manuscript. With funding, we hope to provide MacOS and Linux versions as well. Additionally, a “Getting started” section has been added to the README file which is displayed automatically upon going to the NestIQ GitHub page (https://github.com/wxhawkins/NestIQ).

---

## [Decision Letter · Decision Letter 1]

17 Jul 2020

Applications of machine learning in behavioral ecology: Quantifying avian incubation behavior and nest conditions in relation to environmental temperature

PONE-D-19-30811R1

Dear Dr. DuRant,

We’re pleased to inform you that your manuscript has been judged scientifically suitable for publication and will be formally accepted for publication once it meets all outstanding technical requirements.

Kind regards,

Roland Bouffanais, Ph.D.

Academic Editor

PLOS ONE

Additional Editor Comments (optional):

Reviewers' comments:

Reviewer's Responses to Questions

**Comments to the Author**

1. If the authors have adequately addressed your comments raised in a previous round of review and you feel that this manuscript is now acceptable for publication, you may indicate that here to bypass the “Comments to the Author” section, enter your conflict of interest statement in the “Confidential to Editor” section, and submit your "Accept" recommendation.

Reviewer #1: All comments have been addressed

Reviewer #3: All comments have been addressed

2. Is the manuscript technically sound, and do the data support the conclusions?

Reviewer #1: Yes

Reviewer #3: Yes

3. Has the statistical analysis been performed appropriately and rigorously? 

Reviewer #1: Yes

Reviewer #3: Yes

4. Have the authors made all data underlying the findings in their manuscript fully available?

Reviewer #1: Yes

Reviewer #3: Yes

5. Is the manuscript presented in an intelligible fashion and written in standard English?

Reviewer #1: Yes

Reviewer #3: Yes

6. Review Comments to the Author

Reviewer #1: I have no additional comments. You've addressed what I've asked. I look forward to having a chance to use the program.

Reviewer #3: The authors have substantially improved the manuscript from the first draft. In particular, the changes that they made to the documentation and operation of NestIQ have made it more easily accessible to their target audience of domain specialists who do not want to develop their own custom models for analysis of nesting temperature data.

7. PLOS authors have the option to publish the peer review history of their article (what does this mean?). If published, this will include your full peer review and any attached files.

Reviewer #1: No

Reviewer #3: No

---

## [Editor Report · Acceptance letter]

12 Aug 2020

PONE-D-19-30811R1 

Applications of machine learning in behavioral ecology: Quantifying avian incubation behavior and nest conditions in relation to environmental temperature 

Dear Dr. DuRant:

I'm pleased to inform you that your manuscript has been deemed suitable for publication in PLOS ONE. Congratulations! Your manuscript is now with our production department. 

Kind regards, 

on behalf of

Professor Roland Bouffanais 

Academic Editor

PLOS ONE